# Induction of Stress-Induced Renal Cellular Senescence In Vitro: Impact of Mouse Strain Genetic Diversity

**DOI:** 10.3390/cells10061437

**Published:** 2021-06-08

**Authors:** Chieh Ming Liao, Vera Christine Wulfmeyer, Maxine Swallow, Christine Susanne Falk, Hermann Haller, Ron Korstanje, Anette Melk, Roland Schmitt

**Affiliations:** 1Department of Nephrology and Hypertension, Hannover Medical School, 30625 Hannover, Germany; Liao.chieh@mh-hannover.de (C.M.L.); wulfmeyer.vera@mh-hannover.de (V.C.W.); haller.hermann@mh-hannover.de (H.H.); 2Department of Pediatric Kidney, Liver and Metabolic Diseases, Hannover Medical School, 30625 Hannover, Germany; swallow.maxine@mh-hannover.de (M.S.); Melk.Anette@mh-hannover.de (A.M.); 3Institute of Transplant Immunology, Hannover Medical School, 30625 Hannover, Germany; falk.christine@mh-hannover.de; 4Mount Desert Island Biological Laboratory, Bar Harbor, Maine, ME 04609, USA; 5The Jackson Laboratory for Mammalian Genetics, Bar Harbor, Maine, ME 04609, USA; ron.korstanje@jax.org

**Keywords:** cellular senescence, primary cells, cell culture, genetic background, mouse strains

## Abstract

Cellular senescence, a stress-induced state of irreversible cell cycle arrest, is associated with organ dysfunction and age-related disease. While immortalized cell lines bypass key pathways of senescence, important mechanisms of cellular senescence can be studied in primary cells. Primary tubular epithelial cells (PTEC) derived from mouse kidney are highly susceptible to develop cellular senescence, providing a valuable tool for studying such mechanisms. Here, we tested whether genetic differences between mouse inbred strains have an impact on the development of stress-induced cellular senescence in cultured PTEC. Kidneys from 129S1, B6, NOD, NZO, CAST, and WSB mice were used to isolate PTEC. Cells were monitored for expression of typical senescence markers (SA-β-galactosidase, γ-H2AX+/Ki67−, expression levels of CDKN2A, lamin B1, IL-1a/b, IL-6, G/M-CSF, IFN-g, and KC) at 3 and 10 days after pro-senescent gamma irradiation. Clear differences were found between PTEC from different strains with the highest senescence values for PTEC from WSB mice and the lowest for PTEC from 129S1 mice. PTEC from B6 mice, the most commonly used inbred strain in senescence research, had a senescence score lower than PTEC from WSB and CAST mice but higher than PTEC from NZO and 129S1 mice. These data provide new information regarding the influence of genetic diversity and help explain heterogeneity in existing data. The observed differences should be considered when designing new experiments and will be the basis for further investigation with the goal of identifying candidate loci driving pro- or anti-senescent pathways.

## 1. Introduction

In the past decade, the study of cellular senescence has become an increasingly critical aspect in aging research [1]. Cellular senescence is a stress-induced state of irreversible cell cycle arrest accompanied by a complex phenotype, which promotes inflammation via the senescence-associated secretory phenotype (SASP). Under normal physiological conditions, senescent cells can be removed by the immune system, which plays a role in wound healing and embryonic development. However, with advancing age or in chronic disease, senescent cells accumulate in tissues, where they disturb functional maintenance, promote pathological conditions, and lead to maladaptive repair [2]. 

In the kidney, it has been demonstrated that antagonizing pathways of cellular senescence (e.g., genetic deletion of the cell cycle inhibitor and senescence effector CDKN2A/p16INK4a) leads to an improved outcome after kidney injury [3,4], while an experimental increase in the senescent cell load (e.g., telomere shortening by telomerase knockout) strongly hampers normal repair [5,6]. Several recent studies, in which senescent cells were systemically deleted by transgenic or by pharmaceutical approaches, have indicated that the elimination of senescent cells is efficient in reducing the age-associated loss of renal function and in ameliorating typical histomorphological changes which occur in old kidneys, such as glomerulosclerosis and inflammation [7,8,9]. These findings have led to explorative clinical studies in which the concept of senescent cell elimination is tested in patients with kidney disease [10]. 

Cell-based assays are crucial for a better understanding of the cellular processes and molecular pathways of senescence. Compared to other scientific areas, research on cellular senescence faces the challenge that typical immortalized cell lines cannot be utilized since they bypass important key pathways of senescence. Therefore, several important mechanisms of cellular senescence can only be studied in primary cells. The bulk of the kidney consists of tubular epithelial cells and human and murine primary tubular epithelial cells (PTEC) provide a reliable and convenient in vitro cell model. We and others have previously demonstrated that PTEC are highly susceptible to develop cellular senescence [11,12,13]. Here, we tested the impact of genetic background differences of laboratory and inbred wild-derived mouse strains on the susceptibility of PTEC to enter the stage of cellular senescence.

## 2. Materials and Methods

### 2.1. Cell Culture

Primary tubular epithelial cells (PTEC) were isolated from kidneys of 4-month old male mice as previously described [11,12]. Five strains were included: Classical laboratory strains C57BL/6J, 129S1/SvImJ, NZO/HILtJ (referred to as B6, 129S1, NZO), and inbred wild-derived strains WSB/EiJ, CAST/EiJ (referred to as WSB and CAST). All mice were obtained from the Jackson Laboratory (Bar Harbor, Maine, ME, USA). Kidneys were harvested, minced, and digested in bubble-agitated Hanks 199 medium (Gibco; Thermo Fisher Scientific, Inc., Waltham, MA, USA) containing 0.125% Collagenase Type I (Affymetrix; Thermo Fisher Scientific, Inc., Waltham, MA, USA) at 37 °C for 45 min. The solution was then filtered using a 40 µM cell strainer (BD Biosciences, Heidelberg, Germany) to separate tubules by size. Tubular fragments were cultured in REGM-II medium (Promocell, Heidelberg, Germany) for 6 days until the growing cells reach confluence. Coordinated senescence was induced using 10 grays of γ-irradiation (γ-rays) on day 6 and the cells were passaged one day after irradiation. The development of senescence was subsequently evaluated on days 3 and 10 through senescence markers. For each strain, PTEC were separately extracted from five mice to perform quantitative PCR, immunofluorescence, and SA-β-gal staining. There were three biological replicates of PTEC for each mouse. 

### 2.2. Real Time RT-PCR 

RNA was isolated using NucleoSpin RNA Plus (Machery-Nagel, Düren, Germany). For cDNA synthesis, 1000 ng of RNA was used for reverse transcription by RNA-dependent DNA Polymerase (Promega, Madison, WI, USA). The levels of mRNA expression were determined by TaqMan PCR and quantitative RT-PCR with specific primers (Cdkn2a: forward- 5′-GGG CAC TGC TGG AAG CC-3′, probe- 5′-CCG AAC TCT TTC GGT CGT A-3′, reverse- 5′-AAC GTT GCC CAT CAT CAT C-3′, Lmnb1: forward- 5′-GGG AAG TTT ATT CGC TTG AAG A-3′, reverse- 5′-ATC TCC CAG CCT CCC ATT-3′). For quantitative analysis, relative mRNA levels were calculated according to the 2-ΔΔCt and 2-ΔCt methods for Cdkn2a and Lmnb1, respectively. The Hprt housekeeping gene (forward: 5′-TGA CAC TGG TAA AAC AAT GCA AAC T-3′, reverse- 5′-AAC AAA GTC TGG CCT GTA TCC AA-3′) expression was used to normalize the data.

### 2.3. Senescence-Associated-β-galactosidase (SA-β-gal) Staining

For SA-β-gal staining, cells were fixed by 2% formaldehyde and 0.2% glutaraldehyde in PBS for 10 min. After washing with PBS, the samples were incubated overnight at 37 °C in the staining solution pH 6.0 (40 mM citric acid/Na phosphate buffer, 5 mM K4[Fe(CN)_6_] 3H_2_O, 5 mM K_3_[Fe(CN)_6_],150 mM sodium chloride, 2 mM magnesium chloride, and 1 mg/mL X-gal in distilled water). For each mouse, three wells of PTEC were evaluated and nine random high-power fields (200×) were taken from each well using an Eclipse Ti_2_ microscope (Nikon, Tokyo, Japan). The percentage of positive area was quantified with ImageJ, version 1.52e. 

### 2.4. Immunofluorescence for γ-H2AX/Ki67

PTEC were fixed with 5% paraformaldehyde for 10 min and afterwards permeabilized by 1% Triton in PBS. After blocking with 5% BSA, the samples were subsequently stained with anti-Ki67 antibody (Thermo Scientific, Waltham, MA, USA #14-5698-82) and anti-phospho-histone-H2AX (Cell Signaling, Danvers, MA, USA #9718) antibodies. The primary antibody visualization was achieved using Alexa 488 and Alexa 555 secondary antibodies (Invitrogen, Carlsbad, CA, USA). The DAPI containing mounting medium (Dianova, Hamburg, Germany) was used to counterstain nuclei. For each mouse, three wells of PTEC were evaluated and nine random high-power fields (200×) were taken from each well using an Eclipse Ti2 microscope (Nikon). Cells were counted as senescent when the nuclei presented no Ki67 signal and over five γ-H2AX foci. The positive cells are presented as a percentage. 

### 2.5. Senescence Associated Secretary Phenotype (SASP) Evaluation

Cytokines were quantified using Luminex-based Bioplex mouse Cytokine Panels (Bio-Rad, Hercules, CA, USA) according to the manufacturer’s instructions. The following biomarkers were considered for analysis: Granulocyte colony stimulating factor (G-CSF), granulocyte macrophage colony stimulating factor (GM-CSF), interferon gamma (IFN-γ), keratinocytes-derived chemokine (KC), interleukin-1 alpha (IL-1α), interleukin-1 beta (IL-1β), interleukin-6 (IL-6). Standard curves and concentrations were calculated with the Bio-Plex-Manager 6.2 software (Bio-Rad Laboratories). The seven markers were each included with equal weighting into an integrated SASP value.

### 2.6. Senescence Signature Score

Based on four cellular markers (Cdkn2a, Lmnb1, γ-H2AX+/Ki67−, SA-β-gal) and the integrated SASP value, a combined senescence signature score was calculated for PTEC from each mouse. Values for each marker were normalized according to the following formula:(1)$$\frac{\mathrm{Observed}\;\mathrm{value}}{\mathrm{Average}\;\mathrm{of}\;\mathrm{Observed}\;\mathrm{values}}*100=\mathrm{Normalized}\;\mathrm{values}\;\left(\mathrm{Nv}\right)$$
and a senescence signature score was calculated using the following formula: (2)$$\mathrm{Nv}\left(\mathrm{Cdkn}2a\right)-\mathrm{Nv}\left(\mathrm{Lmnb}1\right)+\mathrm{Nv}\left(\gamma H2\mathrm{AX}\right)+\mathrm{Nv}(\mathrm{SA}\beta \mathrm{gal}+\mathrm{Nv}\left(\mathrm{mean}\;\mathrm{of}\;7\;\mathrm{SASP}\right)\;=\mathrm{senescence}\;\mathrm{signature}\;\mathrm{score}$$

In contrast to the other markers which increase with senescence intensity, Lmnb1 is the only marker that decreases with senescence. In order to obtain a mathematically positive senescence correlation for the combined score, individual Lmnb1 expression values were subtracted from the baseline average.

### 2.7. Statistical Analysis

Results are expressed as means ± SEM. Statistical significance between means for strains was calculated by the one-way ANOVA and Tukey’s multiple comparisons test. Significance between means on days 3 and 10 was calculated by the unpaired *t*-test (GraphPad Software, version 7, San Diego, CA, USA). *p* < 0.05 was considered as statistically significant.

## 3. Results

Kidneys from male mice of the classical laboratory strains C57BL/6J, 129S1/SvImJ, NZO/HILtJ (referred to as B6, 129S1, NZO) and the inbred wild-derived strains WSB/EiJ, CAST/EiJ (referred to as WSB and CAST) were used to generate PTEC (Figure 1). Kidney tissue was digested and tubular fragments were plated onto culture dishes, giving rise to outgrowing epithelial cells. For all mouse strains, cells formed colonies and proliferated vigorously, reaching confluence after 6 days of culture. Subsequently, PTEC were irradiated for coordinated stress-induced cellular senescence induction. Cells were analyzed after passaging and additional 3 and 10 days of culture, using several markers to describe the state of cellular senescence formerly established and used by us and others [11,12,14,15].

### 3.1. SA-β-gal Activity

SA-β-gal activity was low at day 3 (Supplemental Appendix A), but was increased significantly at day 10 in PTEC from all strains (11.9-fold overall increase; *p* = 3.1 × 10^−12^; Figure 2A,B). The highest activity was found in PTEC from WSB mice at both time points, showing significantly more SA-β-gal signal as compared to PTEC from 129S1 and NZO at 3 days and to PTEC from 129S1 at 10 days (*p* = 0.01, *p* = 0.03, *p* = 0.01; Figure 2B).

### 3.2. Cdkn2a Expression

The mRNA expression levels for Cdkn2a (encoding p16INK4a) increased significantly in PTEC from all strains between days 3 and 10 (3-fold overall increase; *p* = 6.6 × 10^−11^; Figure 3A). PTEC from B6 and 129S1 mice had the highest Cdkn2a mRNA expression levels at 3 days (differences reaching statistical significance for B6 versus CAST, B6 versus WSB, 129S1 versus CAST, and 129S1 versus WSB; *p* = 0.01, *p* = 0.007, *p* = 0.03, *p* = 0.02; Figure 3A). At 10 days, Cdkn2a mRNA expression was highest in PTEC from 129S1 mice (difference reaching significance for 129S1 versus WSB; *p* = 0.008; Figure 3A).

### 3.3. Lmnb1 Expression

In contrast to SA-ß-gal activity and Cdkn2a expression, which increase with cellular senescence, Lmnb1 shows an inverse relationship with decreasing levels during the development of senescence. Measuring mRNA expression levels for Lmnb1, we observed a 2.9-fold overall reduction between days 3 and 10 (*p* = 0.003) with a decreased expression in PTEC from all strains (Figure 3B). The strongest decrease in Lmnb1 mRNA expression was found in PTEC from B6 mice, showing the lowest expression level of all strains on day 10 (difference to other strains reaching statistical significance for B6 versus 129S1 and NZO; *p* = 0.004, *p* = 0.005; Figure 3B). The second lowest expression of Lmnb1 mRNA was found in PTEC from CAST mice (difference reaching statistical significance for CAST versus 129S1; *p* = 0.04; Figure 3B). 

### 3.4. Nuclear γ-H2AX Foci 

Another marker for cellular senescence is the occurrence of DNA damage response associated nuclear γ-H2AX in non-proliferating (Ki67-negative) cells. Quantification of γ-H2AX+/ Ki67− cells revealed a 2.9-fold increase from day 3 to 10 over PTEC from all strains (*p* = 2.7 × 10^−10^). Despite some inter-group variations, there were no significant differences between the strains (Figure 4A,B). Of note, we also quantified Ki67-positive cells as a measure of cell proliferation but since we observed no significant differences between strains, Ki67 was not included as a separate marker (data not shown).

### 3.5. SASP Factors

While transcriptomic analyses are valuable for certain senescence factors, they do not directly assess secreted factors of senescent cells [13]. Therefore, we used multiplex peptide analysis to determine cellular concentrations of seven SASP factors IL-1α, IL-1β, IL-6, G-CSF, GM-CSF, IFN-γ, and KC (CXCL1) (Figure 5A). When integrating these factors into a single SASP score, we observed a 2.2-fold increase from day 3 to 10 (*p* = 8.3 × 10^−5^; Figure 5B). Individual differences between the groups only reached statistical significance at day 3 with PTEC from CAST mice showing significantly more SASP factor synthesis as compared to PTEC from 129S1 and NZO mice (*p* = 0.01, *p* = 0.01; Figure 5B).

### 3.6. Senescence Signature Score

Finally, results from all markers were combined into a senescence signature score for each PTEC preparation (individual mice) incorporating respective values for SA-β-gal activity, Cdkn2a expression, Lmnb1 expression, γ-H2AX+/Ki67− cells, and the combined SASP yield. The senescence signature score was calculated for days 3 and 10 separately (Figure 6A) and for the two time points combined (Figure 6B). Overall, there was a 6.3-fold increase in the senescence signature score from day 3 to 10 (*p* = 7.4 × 10^−16^, Figure 6A). When analyzing the two time points separately, PTEC from WSB mice showed the highest senescence signature score with a significantly higher value than PTEC from NZO and 129S1 mice at day 3 (Figure 6A,B). This was similar if the time points were combined, showing again a significantly higher value for WSB compared to NZO and 129S1. Additionally, CAST was significantly higher than 129S1 (Figure 6C). 

## 4. Discussion

Genetic background effects are crucial to experimental research as they modify biological processes, phenotypic characteristics, and disease models. Understanding whether differences in genetic background might influence the development of cellular senescence is lacking. Here, we show that the rate of stress-induced senescence induction in cultured primary kidney cells is different in five mouse strains, indicating an importance of the underlying genetic background. 

While senescent cells display different phenotypical modifications, no single marker can reliably identify or quantify cellular senescence [16]. Therefore, we analyzed different markers individually and used their combination to generate a senescent signature score for each mouse strain. All markers showed significant increases with intensified senescence signaling at 10 compared to 3 days after irradiation. Comparing the respective mouse strains, we observed a heterogeneous intensity and dynamic of the individual senescence markers. For example, in PTEC from 129S1 mice, we observed the highest Cdkn2a expression of all strains but the lowest activity of SA-β-galactosidase. On the other hand, PTEC from WSB mice displayed high SA-β-galactosidase activity but low Cdkn2a expression. In the integrated senescence, signature score 129S1 was the strain with the lowest score, whereas WSB reached the highest score. Differences between individual markers illustrate the complexity of identifying the senescent state and underscore the importance of using a panel of senescence markers to determine senescence. 

Primary cells are indispensable for studying the replicative or stress-induced growth arrest necessary for the induction of molecular senescence, which can only be partially reproduced in immortalized cell lines. An additional advantage of primary cells is that they have not been modified and are therefore thought to reflect the in vivo situation more closely than cell lines. However, it is important to note that the underlying processes of senescence induction may differ between cultured cells and the in vivo situation. According to the concept of a ‘culture shock’ induced senescence [17], we previously found that PTEC develop a spontaneous pro-senescent phenotype during the first days of culture and that an additional stressor (i.e., ionizing irradiation) allows an intensified and coordinated induction of the senescent phenotype [11]. It is not clear how these in vitro stress conditions compare to the ageing or disease-related induction of senescence in vivo. Our data, however, show that cell-intrinsic differences depending on genetic background can have a clear impact on pro-senescent pathways. Future studies will have to address whether similar differences can be shown in vivo.

B6 is the most commonly used inbred strain and most studies on the pathological role of senescence in aging and diseases have used B6 mice. We found that PTEC from B6 mice had a senescence score lower than PTEC from WSB and CAST mice (the two wild-derived strains) but higher than PTEC from the other classical laboratory strains that we tested (NZO and 129S1). Due to its relevance in genetically engineered mice, our findings on PTEC from the B6 strain might be of particular importance. To monitor senescence and to eliminate senescent cells in vitro and in vivo, the Cdkn2a promoter has been used for transgenic or knock-in approaches in B6 mice [7,18,19,20]. Compared to other strains, we show that B6 PTEC develop a high expression of Cdkn2a during senescence induction. Another interesting marker, which has often been used to monitor senescence in organs and cell culture, is the loss of Lmnb1 expression [15]. While we observed high sample variability at day 3 for all strains, the clearest reduction of Lmnb1 expression at day 10 was in PTEC from B6 mice. 

Our data indicate an effect of the underlying genotype for the renal senescence phenotype with the highest combined senescence score for PTEC from WSB and CAST mice. WSB, B6, 129S1, and NZO are primarily from Mus musculus domesticus origin, while CAST is derived from a different subspecies (Mus musculus castaneus). The differences can therefore not simply be ascribed to WSB and CAST being wild-derived strains as opposed to the classical laboratory strains B6, 129S1, and NZO. Similarly, there is no pattern with regards to the median lifespan of the mice: WSB is the longest living strain (median of 952 days for males) and NZO is the shortest (423 days for males) [21]. While we cannot describe a real pattern, due to the small number of strains investigated, we clearly find an effect of genetic diversity on the senescence phenotype. Further studies, using collaborative cross lines [22] or diversity outbred mice [23], which are derived from the strains we used in this study, will be able to identify the genetic factors determining these differences and will allow us to gain further insight into the molecular mechanisms of senescence.

In summary, our results provide new information regarding the influence of genetic diversity, by interrogating primary cultured cells from genetically different mouse inbred strains on the development of senescence. Our results can help explain heterogeneity in existing data and should be considered when designing new experiments. The causal relationship between genetic heterogeneity and the observed differences will be investigated in the future with the goal of identifying candidate loci driving pro- or anti-senescent pathways.

## Figures and Tables

**Figure 1 cells-10-01437-f001:**
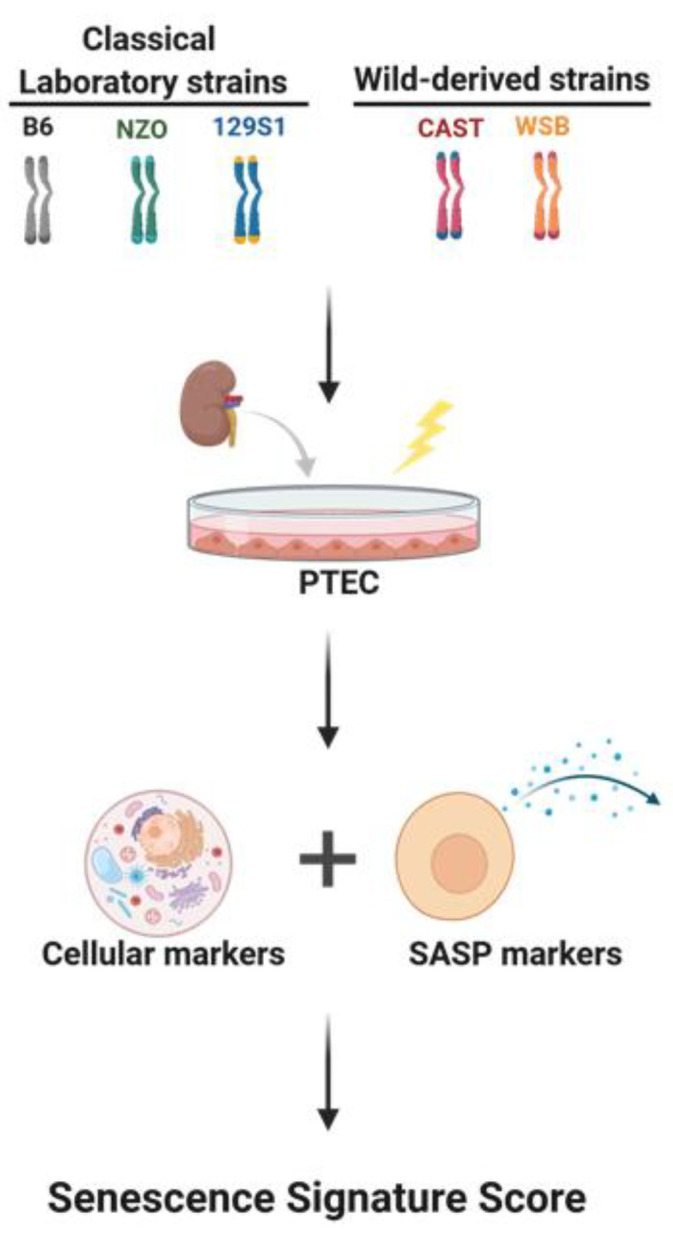
Schematic of experimental procedure and evaluation. Primary tubular epithelial cells (PTEC) generated from five laboratory strains C57BL/6J (B6), 129S1/SvImJ (129S1), NZO/HILtJ (NZO), WSB/EiJ (WSB), and CAST/EiJ (CAST), were irradiated and evaluated 3 and 10 days after irradiation for cellular senescence markers and for secretion of senescence associated secretory phenotype (SASP) markers. All markers were combined to build a senescence signature score for each strain.

**Figure 2 cells-10-01437-f002:**
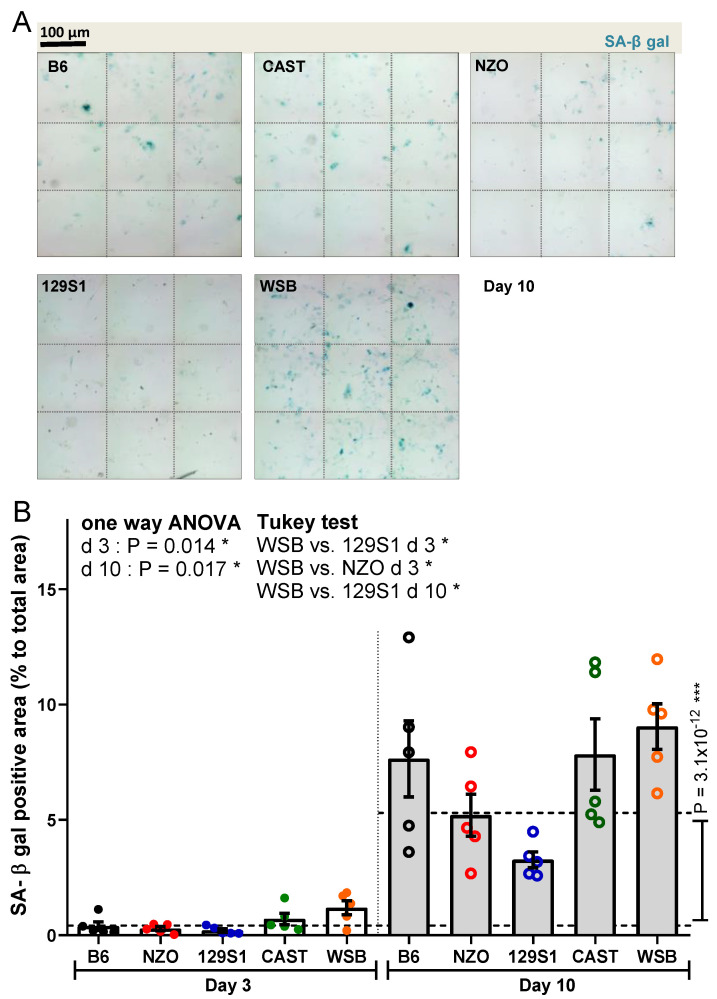
Analysis of cellular senescence using senescence-associated beta-galactosidase (SA-β-gal). (**A**) Representative images of PTEC after SA-β-gal staining (positive cells are blue) at day 10 after irradiation. Original magnification 200 ×. (**B**) Quantification of SA-β-gal staining of PTEC at days 3 and 10 after irradiation. Results are presented as means ± SEM (*n* = 5 mice for each strain). Significance between means of strains and means of time points were tested by the one-way ANOVA and *t*-test, respectively. Tukey’s test was performed as a post-hoc analysis in the case of multiple comparisons. Results from days 3 and 10 are tested separately. * *p* < 0.05; *** *p* < 0.001.

**Figure 3 cells-10-01437-f003:**
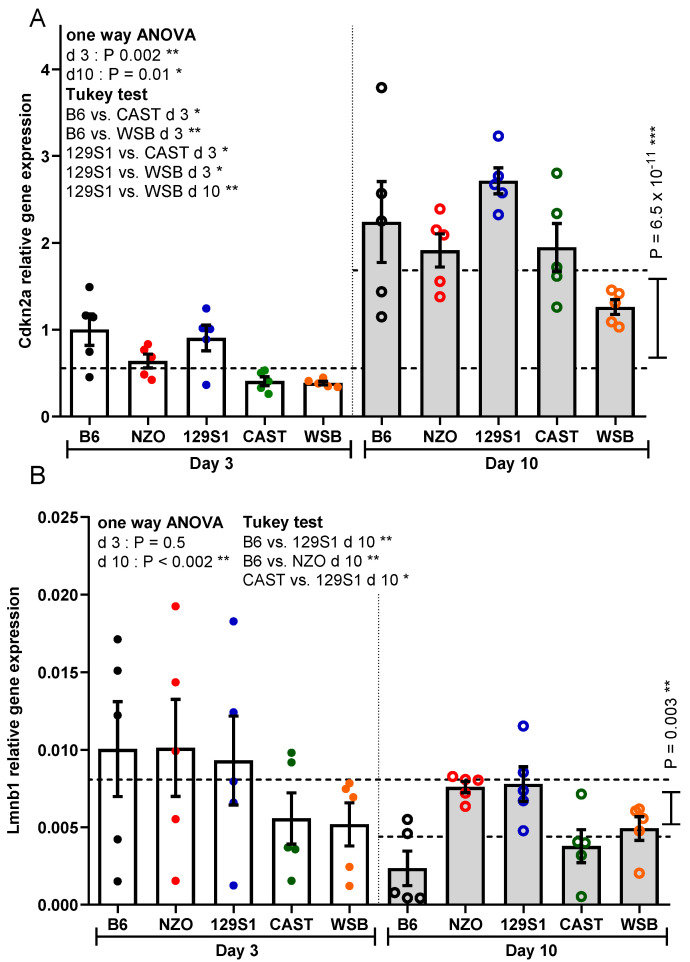
Analysis of cellular senescence using Cdkn2a and Lmnb1 transcriptional expression. (**A**) Cdkn2a expression and (**B**) Lmnb1 expression at days 3 and 10 after irradiation. Results are presented as means ± SEM (*n* = 5 mice for each strain). Significance between means of strains and means of time points were tested by the one-way ANOVA and *t*-test, respectively. Tukey’s test was performed as a post-hoc analysis in the case of multiple comparisons. Results from days 3 and 10 are tested separately. * *p* < 0.05; ** *p* < 0.01; *** *p* < 0.001.

**Figure 4 cells-10-01437-f004:**
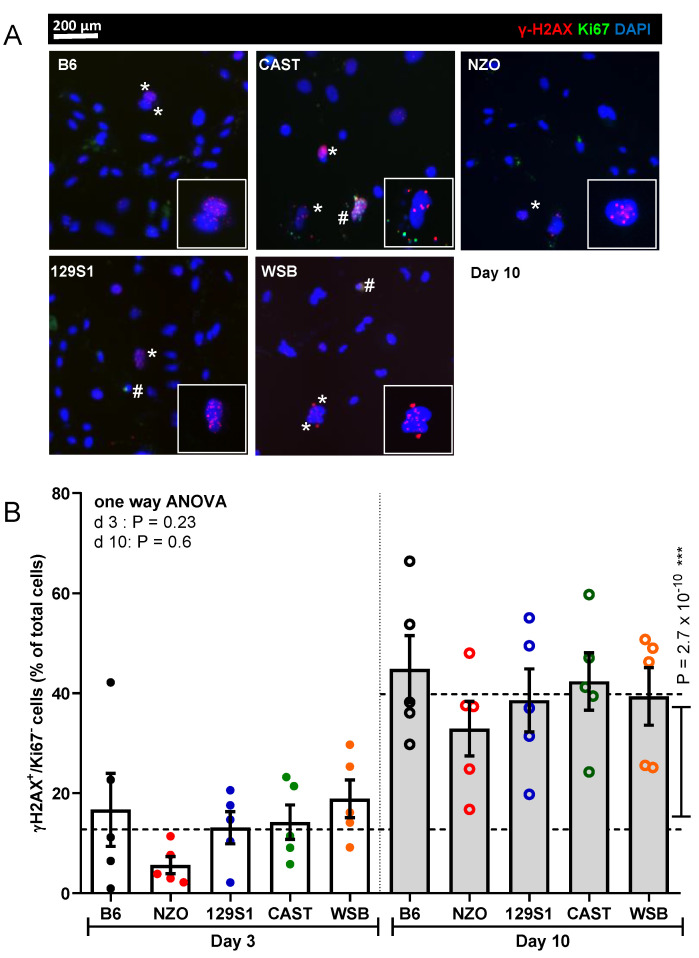
Analysis of cellular senescence using γH2AX and Ki67 immunofluorescence co-staining. (**A**) Representative images of PTEC at day 10 after irradiation showing immunofluorescent staining for γH2AX and Ki67. Cells were counted as senescent (marked with *) when Ki67 negative and simultaneously positive for over five γH2AX foci. Ki67-positive cells are marked with #. Original magnification 400 ×. (**B**) Percentage of senescent cells was quantified at days 3 and 10 after irradiation. Results are presented as means ± SEM (*n* = 5 mice for each strain). Significance between means of strains and means of time points were tested by the one-way ANOVA and *t*-test, respectively. Tukey’s test was performed as a post-hoc analysis in the case of multiple comparisons. Results from days 3 and 10 are tested separately.; *** *p* < 0.001.

**Figure 5 cells-10-01437-f005:**
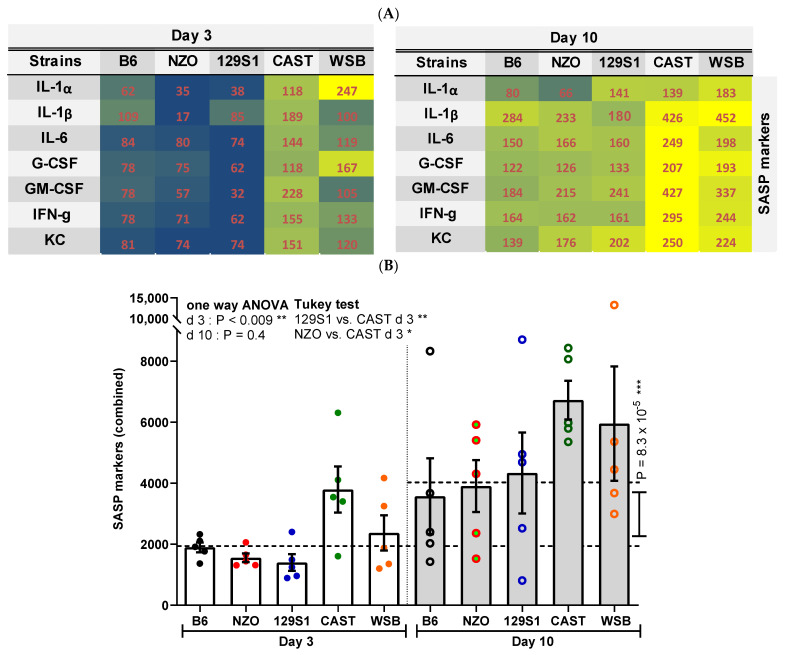
Analysis of cellular senescence using senescence associated secretory phenotype (SASP). (**A**) Protein levels of the indicated seven SASP factors are visualized by heatmap for days 3 and 10 after irradiation. Mean values of the seven SASP factors are depicted for each mouse strain (average normalized to day 3) to define the overall levels of SASP secretion (signal intensities ranked from lowest = blue to highest = bright yellow). (**B**). Corresponding results are presented as means ± SEM (*n*= 5 mice for each strain). Significance between means of strains and means of time points were tested by the one-way ANOVA and *t*-test, respectively. Tukey’s test was performed as a post-hoc analysis in the case of multiple comparisons. Results from days 3 and 10 are tested separately. * *p* < 0.05; ** *p* < 0.01; *** *p* < 0.001.

**Figure 6 cells-10-01437-f006:**
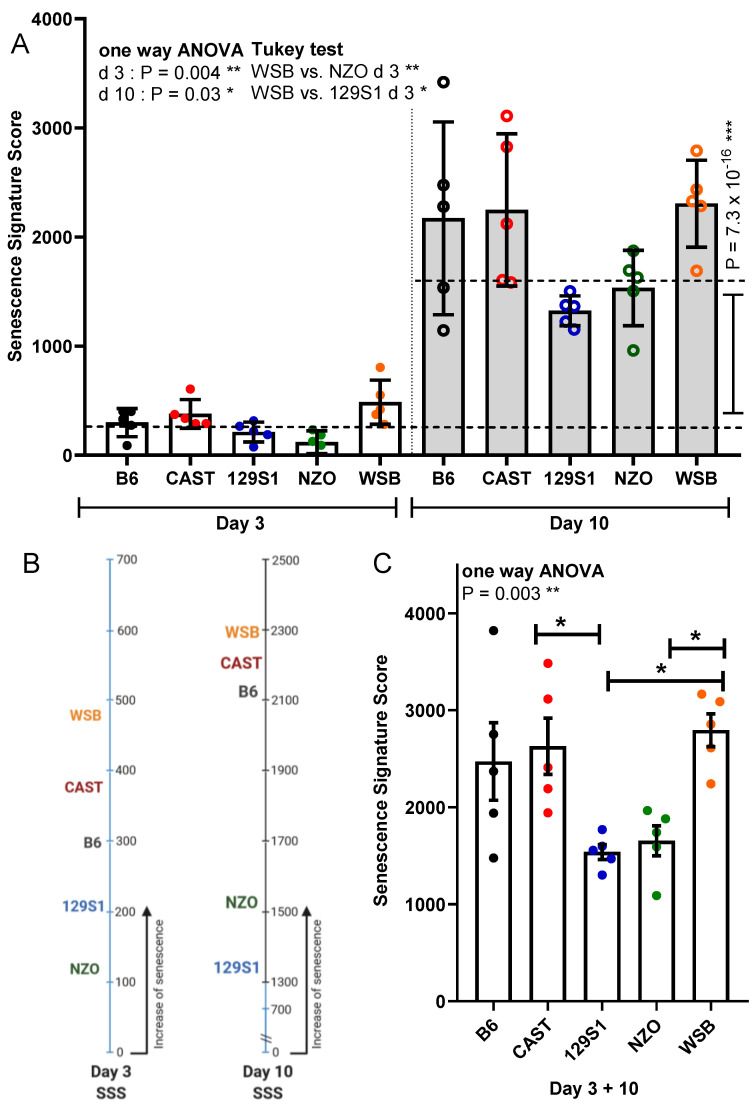
Senescence signature score combining all markers. (**A**) A senescence signature score composed of all five markers (SA-β-gal, γ-H2AX+/Ki67−, Cdkn2a, and Lmnb1 expression, SASP) was calculated for each mouse and categorized by strain to define the overall levels of SASP secretion for days 3 and 10 after irradiation. (**B**) Mean of senescence signature score (SSS) for each strain. (**C**) Combined sum of SSS for days 3 and 10. Results are presented as means ± SEM (*n* = 5 mice for each strain). Significance between means of strains and means of time points were tested by the one-way ANOVA and *t*-test, respectively. Tukey’s test was performed as a post-hoc analysis in the case of multiple comparisons. Results from days 3 and 10 are tested separately. * *p* < 0.05; ** *p* < 0.01; *** *p* < 0.001.

## Data Availability

The data that support the findings of this study are available from the corresponding author upon reasonable request.

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
