# Peer review of "Induction of Stress-Induced Renal Cellular Senescence In Vitro: Impact of Mouse Strain Genetic Diversity"

_cells, 2021, doi:10.3390/cells10061437_

Round 1

Reviewer 1 Report

In this article, the authors investigate the impact of genetic heterogeneity on the induction of senescence in primary tubular epithelial cells (PTEC) derived from different mouse inbred strains. Although the results suggest that, in fact, the genetic background may influence the process of cellular senescence, several concerns must be addressed.

Major comments:

  • The title of the article states that the study focuses on the senescence of tubule cells in vitro. In reality, the authors studied stress-induced premature senescence rather than replicative senescence. To be consistent with the purpose of the study, the authors should have investigated the replicative senescence of PETCs without irradiation. The results obtained in cells undergoing spontaneous replicative senescence would also have provided the appropriate values for the comparisons with the data of stress-induced premature senescence (irradiated cells).
  • The major criterion for defining senescence of proliferative cells is their decreased mitotic properties. The proliferative marker Ki67 has been measured to identify quiescent cells but it has not been quantified for monitoring cell proliferation. Quantification of proliferation of PTEC is missing and must be shown.
  • Besides yH2AX, additional markers of DDR should be measured (ATM/ATR, p53, p21)
  • Fig 5, heatmap: How the heatmap has been defined? What do “low, medium and high” mean? What are the corresponding values? In addition to the heatmap, graphs for each marker would be appreciated. Does the heatmap take into account the variability.
  • Fig 5A: Concerning B6 mice, the heatmap at Day 3 indicates “medium values” for all the markers, while it indicates “low values” at Day 10. Nevertheless, on the quantification on the Fig5B, B6 mice show a slight increase of combined SASP markers at Day 10 compared to Day 3. How the authors explain this discrepancy? .

Author Response

  • The title of the article states that the study focuses on the senescence of tubule cells in vitro. In reality, the authors studied stress-induced premature senescence rather than replicative senescence. To be consistent with the purpose of the study, the authors should have investigated the replicative senescence of PTECs without irradiation. The results obtained in cells undergoing spontaneous replicative senescence would also have provided appropriate values for the comparisons with the data of stress-induced premature senescence (irradiated cells).

We thank the Reviewer for his critical comments and agree that the title of our study can be misleading. Generally, isolated murine PTEC have a high spontaneous tendency to go rapidly into senescence when cultured in vitro. To describe this non-replicative cell-culture stress induced senescence in PTEC we have previously used the term “culture shock senescence” (Baisantry A, et al. 2016. Cell Cycle, 15:21, 2973-2979) which was originally introduced by Sherr and DePinho (Cell. 2000 Aug 18;102(4):407-10). We have also shown previously, that adding a timed pro-senescent stimulus (γ-irradiation) allows for reinforcement and synchronization of the senescence induction process (Berkenkamp B, et al. 2014. PLoS ONE 9(2): e88071). Although we agree with the Reviewer that investigating the pure “culture shock” induced senescence without additional γ-irradiation would have been interesting we chose for the current experiments to use γ-irradiation to allow for improved standardization, timing and synchronizing of the senescence induction process in PTEC. We believe that this was important to minimize intergroup variations and optimized comparability. In the revised version we changed the title to emphasize that we investigated stress-induced senescence and not replicative senescence and changed the manuscript accordingly (title, abstract, p 4, p8).

  • The major criterion for defining senescence of proliferative cells is their decreased mitotic properties. The proliferative marker Ki67 has been measured to identify quiescent cells but it has not been quantified for monitoring cell proliferation. Quantification of proliferation of PTEC is missing and must be shown.

According to the Reviewer’s suggestion we quantified Ki67 positivity as a measure of cell cycling. The percentage for Ki67 positive cells decreased from Day 3 to Day 10 in all strains (review figure 1A in PDF). As we didn’t observe significant strain difference on either time point, however, incorporation of Ki67 positivity into our senescence signature score had no measurable impact on the overall outcome (see review figure 1B in PDF). To keep the score as simple as possible we therefore did not include Ki67 as an additional marker due to lack of additional discrimination. This is mentioned in the revised manuscript (page 6-7).

  • Besides γH2AX, additional markers of DDR should be measured (ATM/ATR, p53, p21)

We agree that testing additional markers might help to solidify findings concerning the DDR pathway. This is particularly true on a whole organ level and in many non-renal cell types. In PTEC however, the overlap between genes up-regulated in DDR and up-regulated in non-DDR related stress is much larger than in other cell types. As expected from our previous unpublished data we observed wide scattering of p21 in the current PTEC experiments with limited total increase between Day 3 and Day 10 (see review figure 2 A in PDF). We know that the culture stress conditions alone already cause a strong induction of p53 and its transactivational target p21, which can be observed rapidly after starting the culture (at a time when we do not reach stable conditions for intergroup comparisons; see review figure 2 B). Our intention was using markers which allow us to discriminate pro-senescent process during our longitudinal follow-up period (Day 3 – Day 10). We focused on broad markers of cellular senescence which we were able to determine with limited sample resources. While the impact of genetic background on the nexus between DDR and senescence is an interesting additional topic, we suggest to address this in future studies with proper experimental design.

  • Fig 5, heatmap: How the heatmap has been defined? What do “low, medium and high” mean? What are the corresponding values? In addition to the heatmap, graphs for each marker would be appreciated. Does the heatmap take into account the variability.

In order to address these questions, we decided to adapt the heatmap design which is now shown in a way which we hope is more self-explanatory. The normalized mean value for each strain can be seen directly in each heatmap field by numerical values. The corresponding color code represents signal intensities ranked from lowest = blue to highest = bright yellow. To illustrate the inter-group variability we included separate graphs for the 7 SASP markers in in the revised manuscript (supplementary Figure 2). The results are variable between mice within the same strain, but significant inter-strain differences can be observed individually for Il-1α, IL-1β and GM-CSF.

  • Fig 5A: Concerning B6 mice, the heatmap at Day 3 indicates “medium values” for all the markers, while it indicates “low values” at Day 10. Nevertheless, on the quantification on the Fig5B, B6 mice show a slight increase of combined SASP markers at Day 10 compared to Day 3. How the authors explain this discrepancy?

We appreciate this comment which has also been made by Reviewer 2. In the original version of the heatmap we used a separate color code range for Day 3 and Day 10. As this might be misleading, we decided to use a time-point independent color code range which allows comparison not only between strains but also between time points (page 8 new Fig 5 A of revised manuscript).

Reviewer 2 Report

Liao et al. present a neat study demonstrating the difference in senescence phenotypes of primary tubular epithelial cells from five different commonly used mouse strains. After inducing senescence by gamma-irradiation, they quantitated five different markers (or sets of markers in the case of SASP) and developed a senescence signature score to incorporate all results. This work demonstrates that the senescence phenotype can differ between models in non-uniform way, and it will be a useful resource to assist researchers in deciding which genetic background they should use to study senescence.

The manuscript is concise and clear and is, in my opinion, acceptable to publish in its current form. I would suggest only the following few suggestions:

  • Is it possible to get some reference for the basal state of some of the markers. For example, are Day 3 levels of Lmnb1 expression significantly lower than in cells without induction of senescence? It would be good to include basal levels on the graphs to get some level of perspective on this.
  • Are the SASP markers in Figure 5A normalized within each day? It looks like the B6 signal is higher at 3 days than at 10 days, however this seems to not be the case in Figure 5B. Would it be better to normalize them all to each other, across Day 3 and Day 10, or would thatt be a misrepresentation of the data?

Author Response

  • Is it possible to get some reference for the basal state of some of the markers. For example, are Day 3 levels of Lmnb1 expression significantly lower than in the cells without induction of senescence? It would be good to include basal levels on the graphs to get some level of perspective on this.

We would like to thank the Reviewer for the encouraging comments and constructive questions. We agree that baseline values for the markers could provide additional information for better illustration of the process of senescence induction in vitro. Unfortunately we are unable to provide baseline values for all strains due to lack of material; however, we show below the induction of markers comparing B6 cells before and at day three and day ten after gamma-irradiation (review figure 3 in PDF). The graphs depict the difference in marker expression including down-regulation of Lmnb1 and up-regulation of γH2AX+/Ki67- cells, SAßGal, p16INK4a and SASP markers (except for G-CSF/Csf3 for which several different sets of primers did not yield reliable qPCR results).

  • Are the SASP markers in Figure 5A normalized within each day? It looks like the B6 signal is higher at 3 days than at 10 days, however this seems to not be the case in Figure 5B. Would it be better to normalize them all to each other, across Day 3 and Day 10, or would that be a misrepresentation of the data?

A similar comment has also been made by Reviewer 1 and we apologized that the set-up and color code of the heatmap was misleading. We now changed the graph to include numerical mean values for each category and used the same color code range for both timepoints. This allows for comparability not only between strains but also between timepoints (page 8, Fig. 5 A of revised version) and creates consistency with the corresponding bar graph (Fig. 5 B).  

Reviewer 3 Report

This original manuscript shows the effect of genetic diversity on renal cellular senescence in vitro. This study is a well-designed experiment using a classical laboratory strain and inbred wild-derived stain. I have minor comments on this work.

  1. In figure 4A, if possible, more larger magnification of IF staining for γ-H2AX and Ki67 might help to understand the DNA damage associated cellular senescence.

Author Response

We would like to thank the Reviewer for the encouraging comments. According to the Reviewer’s suggestion we revised Figure 4 A and chose higher magnification pics and crop-outs for better illustration (page 7 Fig 4 A of the revised manuscript).

Round 2

Reviewer 1 Report

the manuscript has been significantly improved and the responses
to the comments are correct